# TOWARDS HUMAN-LIKE SPOKEN DIALOGUE GENERATION BETWEEN AI AGENTS FROM WRITTEN DIALOGUE

## ABSTRACT

The advent of large language models (LLMs) has made it possible to generate natural written dialogues between two agents. However, generating human-like spoken dialogues from these written dialogues remains challenging. Spoken dialogues have several unique characteristics: they frequently include backchannels and laughter, and the smoothness of turn-taking significantly influences the fluidity of conversation. This study proposes *CHATS* — **CH**atty **A**gents **T**ext-to-**S**peech — a discrete token-based system designed to generate spoken dialogues based on written dialogues. Our system can generate speech for both the speaker side and the listener side simultaneously, using only the transcription from the speaker side, which eliminates the need for transcriptions of backchannels or laughter. Moreover, CHATS facilitates natural turn-taking; it determines the appropriate duration of silence after each utterance in the absence of overlap, and it initiates the generation of overlapping speech based on the phoneme sequence of the next utterance in case of overlap. Experimental evaluations indicate that CHATS outperforms the text-to-speech baseline, producing spoken dialogues that are more interactive and fluid while retaining clarity and intelligibility.

## 1 INTRODUCTION

Large Language Models (LLMs) have profoundly influenced the field of natural language processing (NLP) and artificial intelligence (AI) (Zhao et al., 2023). LLMs, with their capacity to generate coherent and contextually relevant content, have enabled more natural text-based dialogues between humans and computers and paved the way for inter-computer communication. The recently proposed concept of Generative Agents (Park et al., 2023) underscores the potential of LLMs, where emulated agents within the model engage in autonomous dialogues, store information, and initiate actions. This emerging paradigm of agent-to-agent communication offers vast potential across various sectors, from entertainment to facilitating human-to-human information exchange. However, considering the dominance of spoken communication in human interactions, integrating voice into machine dialogues can provide a richer expression of individuality and emotion, offering a more genuine experience. A significant challenge then emerges: how can we transform written dialogues, whether generated by LLMs or humans, into human-like spoken conversations?

Although both written and spoken dialogues serve as mediums for communication, their characteristics and effects on the audience differ significantly. Spoken dialogues are imbued with unique elements such as backchannels, laughter, and smooth transitions between speakers. These are rarely captured fully in written form. For instance, a nod or a simple "uh-huh" serves as a backchannel in spoken dialogues, subtly indicating the listener's engagement and understanding (Yngve, 1970). Similarly, laughter can convey amusement, act as a bridge between topics, and ease potential tensions (Adelswärd, 1989). The smoothness of turn-takings in spoken dialogues, wherein one speaker naturally yields the floor to another, introduces a rhythm and fluidity that is challenging to reproduce in text (Stivers et al., 2009). Several approaches have been proposed to model these backchannels (Kawahara et al., 2016; Lala et al., 2017; Adiba et al., 2021; Lala et al., 2022), laughter (Mori et al., 2019; Tits et al., 2020; Bayramoğlu et al., 2021; Xin et al., 2023; Mori & Kimura, 2023), and turn-taking (Lala et al., 2017; Hara et al., 2018; Sakuma et al., 2023). However, most have focused on human-to-agent conversation or the task itself (e.g., laughter synthesis) and the agent-to-agent situation has not been evaluated.

A straightforward approach for transforming written dialogues into spoken dialogues involves employing a text-to-speech (TTS) system. Advancements in TTS have facilitated the generation of individual utterances at a quality comparable to human voice (Kim et al., 2021; Tan et al., 2022). Certain studies have focused on generating conversational speech by considering linguistic or acoustic contexts (Guo et al., 2021; Cong et al., 2021; Li et al., 2022; Mitsui et al., 2022; Xue et al., 2023). Furthermore, certain studies have equipped LLMs with TTS and automatic speech recognition to facilitate human-to-agent speech communication (Huang et al., 2023; Zhang et al., 2023; Wang et al., 2023; Rubenstein et al., 2023). However, these systems are fully turn-based, where each speaker utters alternatively, and the characteristics of spoken dialogues such as backchannels and turn-taking are neglected. Recently, SoundStorm (Borsos et al., 2023) has succeeded in generating high-quality spoken dialogue; however, it requires transcriptions for backchannels and is subject to a 30-s length constraint. Another approach introduced the dialogue generative spoken language modeling (dGSLM), which generates two-channel audio autoregressively, achieving realistic vocal interactions, laughter generation, and turn-taking (Nguyen et al., 2023). Although dGSLM's operation based solely on audio is revolutionary, it cannot control utterance content via text. Moreover, as reported in section 4.4, generating meaningful content with dGSLM requires a vast dataset.

This study proposes CHATS (**CH**atty **A**gents **T**ext-to-**S**peech), a system for transforming written dialogue into spoken dialogue, whose content is coherent with the input written dialogue but generated with backchannels, laughter, and smooth turn-taking. By conditioning dGSLM on the phonetic transcription of speaker's utterance, our system can generate meaningful and contextually proper utterances on the speaker side. Simultaneously, it generates various backchannels and laughter without transcription on the listener side. The proposed system is designed to overcome the limitations of existing methods, including the turn-based nature of TTS systems and content control constraints of textless models. A collection of audio samples can be accessed through https://anonresearch81.github.io/research/publications/CHATS/.

Our contributions are multi-fold:

- **Exploration of Dual-Tower Transformer Architecture**: Our system is built on top of dGSLM, whose core comprises a dual-tower Transformer to generate discrete acoustic tokens. We condition dGSLM with phonemes and investigate the effect of pre-training in TTS tasks on the textual fidelity. Furthermore, we introduce a pitch representation following Kharitonov et al. (2022) and analyze its effects on both textual fidelity and prosody.
- **Introduction of a Turn-Taking Mechanism**: A novel mechanism for predicting the timing of spoken dialogues is introduced. This encompasses both the duration of pauses after utterances and instances where subsequent utterances overlapped with preceding ones, echoing the organic rhythm and fluidity of human conversations.
- **Extensive Investigation of Generated Spoken Dialogue Characteristics**: We conduct a comprehensive analysis of the characteristics of generated spoken dialogue, comparing its closeness to human dialogue across various dimensions. These include the quality of utterances, the frequency and content of backchannels, the duration of turn-taking events, and the subjective perception of dialogue naturalness.

## 2 BACKGROUND

### 2.1 GENERATIVE SPOKEN LANGUAGE MODELING

Generative Spoken Language Modeling (GSLM) is a framework introduced by Lakhotia et al. (2021) to capture both acoustic and linguistic characteristics of spoken language directly from raw audio, without relying on text or labels. One of the main challenges in raw audio modeling is its excessive information; for instance, a typical audio file contains tens of thousands of samples per second (e.g., 16,000 in 16 kHz audio) and includes various non-linguistic factors like speaker identity and background noise. To effectively process this, GSLM employs a pipelined architecture as shown in Figure 1. The first step involves encoding the raw audio into a sequence of discrete *Units*. This encoding aims to reduce information density (as units are typically at 50 Hz) and to discard non-linguistic information. These units are automatically discovered by clustering the hidden features of a pre-trained self-supervised learning (SSL) model. The module responsible for this conversion is collectively referred to as the speech-to-unit (s2u) module. Subsequently, a unit Language Model

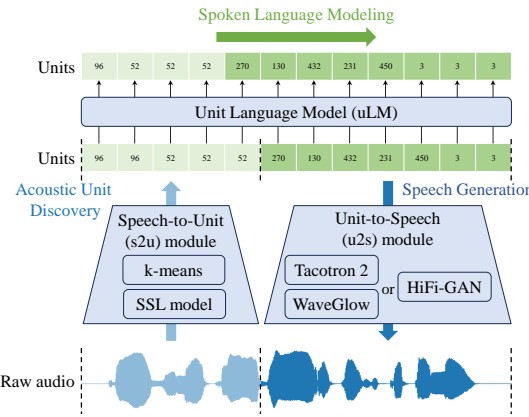

Figure 1: Overview of GSLM pipeline

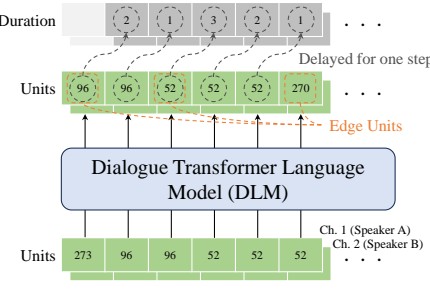

Figure 2: DLM architecture

(uLM) is trained on these discrete units. Similar to language models used for NLP, the uLM employs the standard Transformer architecture and can generate continuations of existing unit sequences once it has been trained. The final step involves the unit-to-speech (u2s) module, which transforms these units back into raw audio. Originally, the u2s module combined a TTS model, Tacotron 2 (Shen et al., 2018), with a neural vocoder, WaveGlow (Prenger et al., 2019). However, recent studies (Kharitonov et al., 2022; Nguyen et al., 2023) have replaced these with a single neural vocoder, HiFi-GAN (Kong et al., 2020).

## 2.2 DIALOGUE GENERATIVE SPOKEN LANGUAGE MODELING

Nguyen et al. (2023) applied the GSLM framework to model spoken dialogues directly, wherein two speakers' voices were recorded separately in two-channel audio. This framework is referred to as dialogue Generative Spoken Language Modeling (dGSLM). While the s2u and u2s modules were remained similar to the original GSLM, a novel architecture for uLM called Dialogue Transformer Language Model (DLM) was proposed to handle two channels of units simultaneously, as illustrated in Figure 2. DLM comprises two towers of Transformers that share their weights, allowing for interactions between two-channel units. In addition, DLM introduces *Edge Unit Prediction* and *Delayed Duration Prediction* objectives to efficiently model the repeating units (e.g. 96, 96, 52, 52, 52, . . . ). The edge unit prediction forces the model to predict the next unit only if it differs from the current one (i.e. edge unit). The delayed duration prediction allows the model to predict the duration of an edge unit at time step $t$ with a one-step delay (i.e. at time step $t + 1$).

## 3 CHATS

### 3.1 SYSTEM ARCHITECTURE

Our system aims to generate spoken dialogues wherein the spoken content aligns with input written dialogues but listener's responses (e.g. backchannels and laughter) are automatically generated. To address this challenging task, we adopt the pipeline architecture of GSLM (Lakhotia et al., 2021), comprising three primary modules: s2u module, uLM, and u2s module.

### 3.1.1 SPEECH-TO-UNIT (S2U) MODULE

The s2u module extracts a concise representation from speech signals, operating on the entirety of a spoken dialogue. It (1) facilitates easy modeling by the uLM and (2) retains the necessary detail for the u2s module to reconstruct a high-fidelity waveform. Following Kharitonov et al. (2022), our s2u module extracts two distinct representations: *Content Units* and *Pitch Units*. The content units, which are identical to the "units" described in section 2, are used to capture the spoken content information. It is obtained using a combination of a pre-trained Hidden-Unit BERT (HuBERT) (Hsu et al., 2021) and a k-means clustering (MacQueen, 1967). The pitch units are used to capture the prosody of speech, which is often discarded in content units. It is obtained by quantizing the speaker-

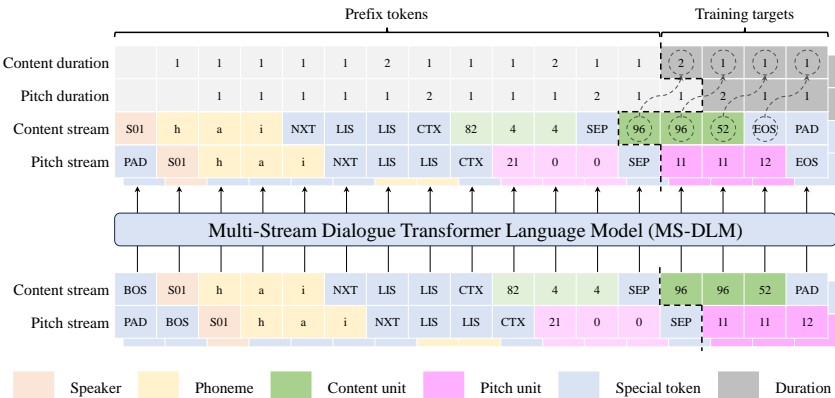

Figure 3: Overview of MS-DLM. It processes content and pitch streams to predict subsequent units and their delayed durations. Each stream comprises a speaker ID, phonemes of current and next utterances, context units, and units to be generated. Each channel corresponds to a different speaker. Phonemes are replaced with listening (LIS) tokens when the utterance is made by the other speaker.

normalized logarithm of the fundamental frequency ($\log F_0$). For the notation, these units are referred to as $u_{n,t}^{c,k}$ or simply $u_t^{c,k}$ when the $n$th utterance need not be highlighted, where $n$ is the utterance index, $t$ is the timestep, $c$ is the audio channel, and $k$ is the codebook index associated with the content and pitch units, respectively. We assume $c, k \in \{1, 2\}$ in this study.

### 3.1.2 UNIT LANGUAGE MODEL (uLM)

The uLM is designed to generate content and pitch units for two channels based on written dialogue. In contrast to s2u and u2s modules, the uLM focuses on individual utterances, rather than entire dialogues, owing to inherent sequence length limitations. However, our uLM only requires the text of the current and next utterances to generate the current speech, thus facilitating sequential production of spoken dialogues without waiting for the generation of the entire written dialogue.

**Model Architecture:** The uLM architecture is based on the DLM (Nguyen et al., 2023) described in section 2.2, which comprises two decoder-only Transformer towers that share parameters. We propose a novel *Multi-Stream DLM (MS-DLM)* architecture for handling multiple streams. We extend the DLM to include two input and output projection layers associated with the *content* and *pitch streams*, respectively, wherein the content and pitch unit sequences are prefixed with the tokens described in the subsequent paragraph. The detailed architecture is depicted in appendix A.1.

**Prefix tokens:** We meticulously design the input sequences of our uLM to facilitate the text-based control over spoken content. The proposed sequences, as illustrated in Figure 3, are as follows:

$$\text{BOS}, s^c, p_{n,1}^c, \ldots, p_{n,M_n}^c, \text{NXT}, p_{n+1,1}^c, \ldots, p_{n+1,M_{n+1}}^c, \text{CTX}, u_{t-C}^{c,k}, \ldots, u_{t-1}^{c,k}, \text{SEP} \quad (1)$$

where $s^c$ is the speaker ID of channel $c$, $M_n$ is the number of phonemes in the $n$th utterance, $C$ is the predetermined context length, and $p_{n,m}^c$ is the $m$th phoneme of the $n$th utterance if uttered by speaker $s^c$, and otherwise substituted with listening (LIS) token. BOS, NXT, CTX, SEP tokens represent beginning of sentence, phonemes of the next utterance, context units, and separator, respectively. Building on the practices from Kharitonov et al. (2022), the uLM delays the pitch stream by one step considering their high correlation with content stream. Positions without tokens owing to this delay are filled with padding (PAD) tokens. Additionally, the target sequence obtained by shifting the input sequence by one step is appended with an end-of-sentence (EOS) token.

The conditioning of the uLM on the speaker ID compensates for the context length constraint, ensuring that the model retains each speaker's unique characteristics. Further, phonemes of the $n+1$th utterance are essential for handling overlaps, particularly if the $n+1$th utterance disrupts the $n$th one. With these prefix tokens, our uLM generates speaker's unit sequences from phonemes conditionally, and listener's unit sequences (may contain backchannels and laughter) unconditionally.

**Training Objective:** The model adopts both the edge unit prediction and delayed duration prediction techniques, proposed by Nguyen et al. (2023), for both content and pitch streams. Full details can be found in appendix A.2.

| | | |
|---|---|---|
| 0.000 | 1.500 | A: Hey, thinking of seeing that new movie this weekend. |
| 1.800 | 3.000 | B: "Time's Mirage"? |
| 3.300 | 5.000 | A: Yeah, that one. Coworker said it's good. |
| 5.000 | 5.300 | B: Uh-huh. |
| 5.100 | 6.500 | A: Mentioned something about great visuals. |
| 7.300 | 8.000 | B: And the music? |
| 8.200 | 10.100 | A: Right! They loved the soundtrack. Made them dance in their seat, apparently. |
| 9.400 | 10.200 | B: Hahaha! |
| 10.500 | 12.000 | B: Sounds fun. Let's go together. |

A: Hey, thinking of seeing that new movie this weekend.
B: "Time's Mirage"?
A: Yeah, that one. Coworker said it's good. Mentioned something about great visuals.

B: And the music?
A: Right! They loved the soundtrack. Made them dance in their seat, apparently.

B: Sounds fun. Let's go together.

(a) Spoken dialogue transcription      (b) Written dialogue

Figure 4: Comparison of (a) raw spoken dialogue transcription and (b) typical written dialogue.

### 3.1.3 UNIT-TO-SPEECH (U2S) MODULE

The u2s module is developed to solve an inverse problem of s2u module. It is trained to reconstruct the original waveform given content and pitch units extracted using the s2u module. As content and pitch units contain minimal speaker information, the u2s module also accepts a speaker embedding. Following Kharitonov et al. (2022), we adapt the discrete unit-based HiFi-GAN (Polyak et al., 2021).

## 3.2 PREPROCESSING AND MODELING TECHNIQUES FOR SPOKEN DIALOGUE

### 3.2.1 WRITTEN DIALOGUE PREPARATION VIA BACKCHANNEL EXCLUSION

We consider a dataset comprising recordings of spontaneous dialogues between two speakers, each accompanied by its transcription (Figure 4 (a)). These transcriptions inherently contain elements not usually present in standard written dialogues (Figure 4 (b)), such as timestamps and listener responses, including backchannels and laughter. Training CHATS directly on these raw transcriptions would be suboptimal, as the system might then fail to replicate these spontaneous behaviors when processing typical written dialogue inputs. To address this, we remove elements like backchannels and laughter from the transcriptions using a combination of rule-based and machine learning approaches. This modification ensures that the system learns to autonomously generate these behaviors in the listener's responses.

First, we omit the temporal metadata and retain only the verbal content. In this process, successive utterances from an identical speaker are merged if they are separated by a silence of $< 200$ ms, and are referred to as inter-pausal units (IPUs). Subsequently, we remove the listener's IPUs, which correspond to backchannels and laughter, from the transcription through the following steps:

**Step 1** If one speaker's IPU encompasses another's, it is termed the *speaker IPU* (*s-IPU*), while the latter is termed the *listener IPU* (*l-IPU*). Any IPUs not fitting these definitions are labeled as *undefined IPU*s (*u-IPU*s).

**Step 2** A binary classifier , hereinafter referred to as *IPU classifier*, is trained to determine whether a given IPU is an *s-IPU* or *l-IPU* based on its content units. The training is conducted using speech segments corresponding to *s-IPU*s and *l-IPU*s identified in step 1.

**Step 3** The classifier trained in step 2 is then applied to categorize the *u-IPU*s.

**Step 4** IPUs identified as *l-IPU*s in steps 1 or 3 are excluded from the transcription.

Consequently, the resulting written dialogues are composed exclusively of *s-IPU*s. Hereinafter, "utterance" denotes an *s-IPU* unless otherwise specified.

### 3.2.2 TURN-TAKING MECHANISM (TTM)

To simulate natural turn-taking, which includes overlapping speech, the uLM is trained using a simple and effective approach. Considering two successive utterances, turn-taking can be bifurcated into two scenarios: *no overlap* and *overlap*. These are shown in the top section of Figure 5. Let $a_n$ and $b_n$ be the start and end times of the $n$th utterance, respectively. The conditions for *no overlap* and *overlap* can be described by $b_n \leq a_{n+1}$ and $b_n > a_{n+1}$, respectively. These start and end times are modified as follows:

$$\hat{b}_n = \hat{a}_{n+1} = \max(b_n, a_{n+1}) = \begin{cases} b_n & \textit{(overlap)} \\ a_{n+1} & \textit{(no overlap)} \end{cases} . \tag{2}$$

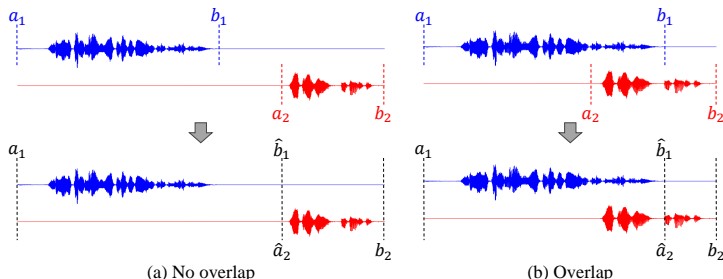

(a) No overlap                         (b) Overlap

Figure 5: Two scenarios of turn-taking, (a) *no overlap* and (b) *overlap*.

The modified time boundaries are shown in the bottom section of Figure 5. Following these alterations, our uLM is trained to predict the duration of trailing silence in the *no overlap* scenario, and pinpoint the onset of overlap in the *overlap* scenario. In the *Overlap* scenario, the uLM must generate the first $b_n - a_{n+1}$ seconds of the $n+1$th utterance concurrently with the $n$th utterance; thus we condition our uLM with the phonemes of the $n+1$th utterance. Moreover, the uLM is tasked with the continuation of the $n+1$th utterance in the *overlap* scenario, justifying our decision to condition the uLM using context units.

### 3.2.3 DATA AUGMENTATION BY CONTEXT REDUCTION

Although context units are included in the prefix tokens, they are not available during the initial steps of inference, which leads to suboptimal generation quality at the start of the dialogue. To address this, data augmentation is proposed, wherein the context is either removed or shortened. We augment the dataset by modifying the context length to $C' = \{0, 0.1C, 0.2C, ..., 0.9C\}$ for each training example. This augmentation is only performed for utterances that do not overlap with previous utterances, as the uLM must generate continuations of context units in the *overlap* scenario.

### 3.3 INFERENCE PROCEDURE

Considering a written dialogue comprising $N$ utterances and speaker pair information $(s^1, s^2)$, a corresponding spoken dialogue can be generated as follows. For each utterance indexed by $n = 1, \ldots, N$, first, the prefix tokens are acquired. The phonemes of the $n$th and $n+1$th utterances are derived using a grapheme-to-phoneme tool, while the context units are sourced from the units generated in previous steps. Then, the content and pitch units of the $n$th utterance are generated autoregressively using the uLM. The process concludes when the `EOS` token is chosen as the content unit for any channel. Thereafter, the delayed pitch units are synchronized with the content units and concatenated to the units that were produced in the earlier steps. Subsequently, the two desired waveform channels are derived using the u2s module. Notably, since our system does not rely on input sentences that extend beyond two sentences ahead, it can facilitate continuous spoken dialogue generation when integrated with an LLM. Illustrative explanation is provided in appendix A.3.

## 4 EXPERIMENTS

### 4.1 SETUP

**Datasets:** We used internal spoken dialogue dataset comprising 74 h of two-channel speech signals (equivalent to 147 h of single-channel speech signals). It includes 538 dialogues conducted by 32 pairs with 54 Japanese speakers (certain speakers appeared in multiple pairs) with their transcriptions. Additionally, we utilized the Corpus of Spontaneous Japanese (CSJ) (Maekawa, 2003) to pre-train our uLM. It contains single-channel speech signals with their phoneme-level transcriptions. All of these were utilized, excluding dialogue data, resulting in 523 h from 3,244 speakers. A detail of our internal dataset and complete procedure of preprocessing are described in appendix B.1.

**Model, training, and inference:** A simple 3-layer bidirectional LSTM was used for the IPU classifier described in section 3.2.1. For the s2u module, we utilized a pre-trained japanese-hubert-base[1] model for content unit extraction, and the WORLD vocoder (Morise et al., 2016) for pitch

---

[1]https://huggingface.co/rinna/japanese-hubert-base

unit extraction. For the uLM model, a Transformer model comprising 6 layers, 4 of which were cross-attention layers, with 8 attention heads per layer and an embedding size of 512 was considered (Nguyen et al., 2023). This uLM was developed atop the DLM implementation found in the fairseq library[2] (Ott et al., 2019). A single-channel variant of our uLM was pre-trained on the CSJ dataset. Subsequently, we finetuned a two-channel uLM on all of the *s-IPU*s from our spoken dialogue dataset. Model optimization was performed over 100k steps on two A100 80GB GPUs with a batch size of 30k tokens per GPU, requiring approximately 5 h for pre-training and 11 h for finetuning. During inference, nucleus sampling (Holtzman et al., 2020) with $p = 0.9$ was adopted. The u2s module utilized the discrete unit-based HiFi-GAN (Kong et al., 2020; Polyak et al., 2021) with minor adjustments. This model was optimized over 500k steps on a single A100 80GB GPU with a batch size of 16 0.5-second speech segments, requiring approximately 32 h. Further details are provided in appendix B.2.

## 4.2 UTTERANCE-LEVEL EVALUATION

First, we focused on the utterance-level generation quality of the proposed system. The fidelity of the generated speech to the input text was investigated by evaluating our system in the TTS setting. We generated speech waveform corresponding to all 4,896 utterances in the test set separately and measured their phoneme error rate (PER). To perform phoneme recognition, we finetuned japanese-hubert-base model with the CSJ dataset. We compared the performance of the proposed system (*Proposed*) with other systems, including 1) *Ground Truth*, the ground-truth recordings, 2) *Resynthesized*, where we combined s2u and u2s modules to resynthesize the original waveform, and 3) *Baseline*, a single-channel counterpart of *Proposed* trained without phonemes of next sentence and the turn-taking mechanism. Additionally, we ablated several components including pre-training on CSJ dataset (*w/o pre-training*), data augmentation by context reduction (*w/o augmentation*), context units (*w/o context*), and phonemes of next sentence

Table 1: PER measured in TTS setting. The lowest PER in each section are bolded.

| METHOD | PER $\downarrow$ |
|---|---|
| *Ground Truth* | 8.95 |
| *Resynthesized* | 11.49 |
| *Baseline* | **12.13** |
|   *w/o pretraining* | 14.10 |
| *Proposed* | 13.03 |
|   *w/o pretraining* | 15.32 |
|   *w/o augmentation* | 59.35 |
|   *w/o context units* | 14.12 |
|   *w/o next sentence* | **12.79** |

(*w/o next sentence*). PERs for *Ground Truth* and *Resynthesized* include both grapheme-to-phoneme error and phoneme recognition error, while *Baseline* and *Proposed* include only the latter.

The results are summarized in Table 1. Although the PER for the *Proposed* system was slightly worse than for *Baseline*, the degradation was minute considering that it performed other tasks in addition to basic TTS, including generating the listener's speech and predicting turn-taking. Pre-training and use of the context units were effective, and data augmentation was crucial because no context was given in the TTS setting. The *Proposed w/o next sentence* marginally outperformed *Proposed* in TTS setting; however, it often generated unnatural or meaningless content as overlapping segment. We investigated the effect of introducing pitch units in appendix C.

## 4.3 DIALOGUE-LEVEL EVALUATION

Next, we evaluated the spoken dialogue generation quality of the proposed system. We quantified how close the generated spoken dialogues were to the recorded ones from two aspects: listener's and turn-taking events. For comparison, we prepared a *Baseline* system, the same system described in section 4.2 but operated alternatively to generate spoken dialogue, as well as *dGSLM* (Nguyen et al., 2023). As *Baseline* cannot generate the listener's tokens, we filled them with the most frequently used content and pitch units corresponding to unvoiced frames. Furthermore, *Proposed w/o TTM* was evaluated to investigate the effectiveness of our turn-taking mechanism.

We created written dialogues that excluded listener's events for the test set as detailed in section 3.2.1. Next, we generated the entire spoken dialogues from those written dialogues. For *dGSLM*, we utilized 30 s of speech prompts from the test set to generate the subsequent 90 s. As the resulting dialogues for *dGSLM* were three times longer than the original test set, we divided the results (e.g., backchannel frequency and duration) by three.

---

[2]https://github.com/facebookresearch/fairseq

### 4.3.1 Listener's event evaluation

We applied the Silero Voice Activity Detector (VAD)[3] to the generated spoken dialogues and performed hybrid IPU classification for each IPU as in section 3.2.1. We then counted the number of backchannels $q_{BC}$ and all utterances $q_{ALL}$ along with their durations $d_{BC}$ and $d_{ALL}$. The results are summarized in Table 2. Although the backchannel frequency and duration for *Proposed* were lower than for *Ground Truth*, the proportion of backchannels in all utterances was closest to the *Ground Truth* in terms of both frequency and duration. *dGSLM* tended to produce too many backchannels, whereas *Baseline* produced too few. Further, *Proposed w/o TTM* produced excessive backchannels. We conjecture that the uLM generates overlapped segments twice without the TTM (as the last part of the $n$th utterance and the first part of the $n + 1$th utterance), resulting in unwanted backchannels. Further investigation of backchannel content and speaker-specific characteristics, as well as laughter frequency and duration, is described in appendix D.

Table 2: Backchannel frequency $q$ and duration $d$. Ratios closest to the *Ground Truth* are bolded.

| METHOD | $q_{BC}$ | $q_{ALL}$ | $100 \times q_{BC}/q_{ALL}$ | $d_{BC}$ [s] | $d_{ALL}$ [s] | $100 \times d_{BC}/d_{ALL}$ |
|---|---|---|---|---|---|---|
| *Ground Truth* | 1854 | 9453 | 19.61 | 1518 | 16588 | 9.15 |
| *dGSLM* | 1710 | 6141 | 27.84 | 1678 | 12378 | 13.56 |
| *Baseline* | 76 | 3656 | 2.08 | 151 | 11713 | 1.29 |
| *Proposed* | 1535 | 6668 | **23.02** | 1322 | 14001 | **9.44** |
| *w/o TTM* | 1756 | 5273 | 33.30 | 1480 | 14052 | 10.53 |

### 4.3.2 Turn-taking event evaluation

Following Nguyen et al. (2023), we examined the distribution of four turn-taking events: 1) *IPU*, a speech segment in one speaker's channel delimited by a VAD silence of $\geq 200$ ms on both sides, 2) *overlap*, a section with voice signals on both channels, 3) *pause*, a silence segment between two IPUs of the same speaker, and 4) *gap*, a silence segment between two IPUs by distinct speakers. The results are summarized in Figure 6. Both *dGSLM* and *Proposed* exhibited similar distribution to the *Ground Truth*, confirming that the proposed system could mimic human-like turn-taking. The distribution of *Baseline*, particularly for overlaps, deviated significantly from that of the *Ground Truth* because theoretically it cannot generate any overlaps. The durations of pauses and gaps were underestimated for *Proposed w/o TTM*, which is congruent with the idea that the TTM is helpful for estimating appropriate silence durations following each utterance. Speaker-specific characteristics of turn-taking events are investigated in appendix E.

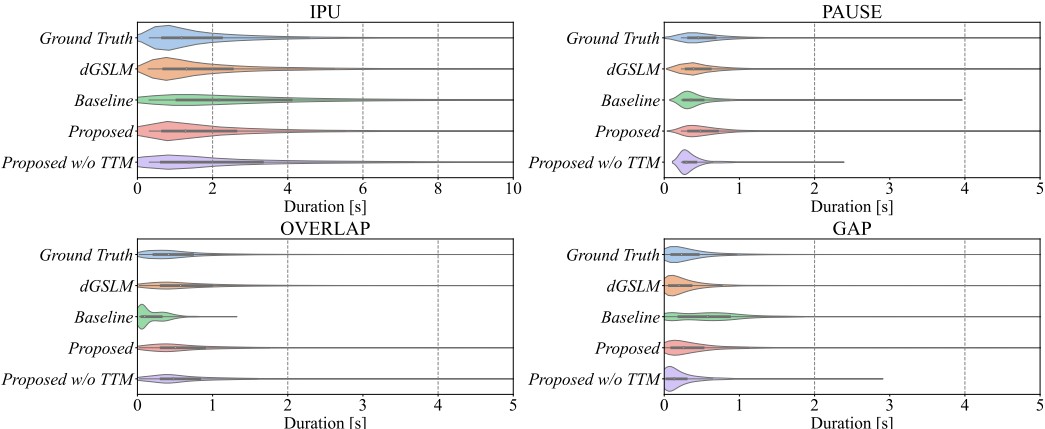

Figure 6: Distributions of turn-taking event durations.

---

[3]https://github.com/snakers4/silero-vad

## 4.4 HUMAN EVALUATION

Finally, we measured the subjective quality of the generated spoken dialogue. For each speaker pair, we randomly extracted two 10-turn dialogues, each lasting 15–45 seconds, from the test set, leading to a total of 64 dialogues. We generated the corresponding spoken dialogue segments using the *Baseline* and *Proposed* systems. For *dGSLM*, we used 30 s of the recorded speech segments preceding these dialogues as prompts and generated 30 s continuations for each one. Each dialogue segment was assessed based on three distinct criteria: 1) *Dialogue Naturalness*, evaluating the fluidity of the dialogue and the naturalness of the interaction, 2) *Meaningfulness*, determining the comprehensibility of what is spoken, and 3) *Sound Quality*, checking for noise or distortion in the speech signal. Each item was rated on a 5-point scale from 1–5 (bad to excellent). Twenty-four workers participated in the evaluation and each rated 25 samples. The instructions and dialogue examples actually used for the evaluation are presented in appendices F and G, respectively.

The results are presented in Table 3. The *Proposed* system outscored both the *dGSLM* and *Baseline* systems across all metrics. Particularly, it recorded a significantly higher score in Dialogue Naturalness compared to the *Baseline* system ($p = 0.038$ in the Student's t-test). Thus, features such as backchannels, laughter, and seamless turn-taking, rendered possible by the proposed system, are vital for generating natural spoken dialogues. Interestingly, *dGSLM* had low scores in both Meaningfulness and Dialogue Naturalness. This finding is at odds with the results from a previous study (Nguyen et al., 2023). We hypothesize that this decline in performance was owing to the smaller dataset used (2,000 h in the previous study vs. 74 h in this study). However, considering that Meaningfulness of *dGSLM* was low in the previous study as well, our system's text conditioning capability proves to be highly effective for generating meaningful spoken dialogue.

While our findings indicate advancements in spoken dialogue generation, certain areas require further refinement to match human-level performance. Notably, the Sound Quality of the *Resynthesized* is behind that of the *Ground Truth*, suggesting the necessity for improved s2u and u2s modules with enhanced speech coding. Moreover, the *Proposed* system trails in Dialogue Naturalness when compared to both the *Ground Truth* and *Resynthesized*. Thus, our future efforts will focus on accumulating a more extensive dialogue dataset and refining our method accordingly.

Table 3: Human evaluation results.

| METHOD | Dialogue Naturalness | Meaningfulness | Sound Quality |
|---|---|---|---|
| *Ground Truth* | 4.85±0.08 | 4.81±0.09 | 4.75±0.09 |
| *Resynthesized* | 4.48±0.12 | 4.55±0.12 | 3.82±0.18 |
| *dGSLM* | 2.68±0.24 | 1.18±0.07 | 2.93±0.20 |
| *Baseline* | 3.01±0.20 | 3.43±0.18 | 3.22±0.18 |
| *Proposed* | **3.30±0.18** | **3.58±0.17** | **3.38±0.18** |

## 5 CONCLUSION

This study proposed CHATS, a system that generates spoken dialogues from written ones. We proposed conditioning uLM with speaker, text, and past speech to achieve coherent spoken dialogue. Additionally, we proposed a mechanism for handling the timing for turn-taking or speech continuation explicitly. We performed a detailed analysis on the generated spoken dialogue, which showed that the proposed system reproduced the ground-truth distribution of backchannel frequency and turn-taking event durations well. Further, the results of our human evaluations demonstrated that the proposed system produced more natural dialogue than the baseline system, which used a TTS model to generate spoken dialogue. We verified that the innovative capability of the proposed system to generate backchannels and laughter without transcriptions was effective in mimicking human dialogue and creating natural spoken dialogue. However, there is still ample room for improvement. To further bridge the divide between human and generated dialogues, we plan to expand our study to a larger dataset for better naturalness and sound quality. Additionally, we will explore the advantages of conditioning our model on raw text to better understand the context of written dialogues. Furthermore, evaluating our system from the aspect of speaking style consistency and expressiveness is a valuable research direction.

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

# A MS-DLM DETAILS

## A.1 MODEL ARCHITECTURE

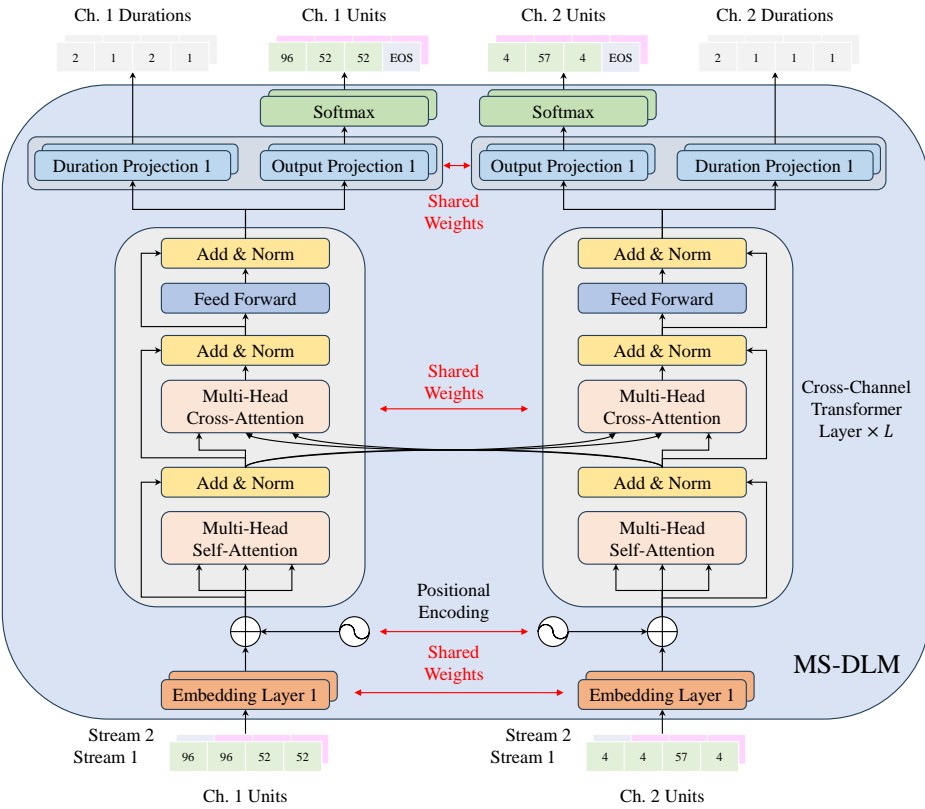

Figure A.1: MS-DLM architecture. All weights are shared across two Transformer towers.

## A.2 TRAINING OBJECTIVE

MS-DLM predicts the unit $u_{n,t}^{c,k}$ and its duration $d_{n,t}^{c,k}$ only when $u_{n,t}^{c,k} \neq u_{n,t-1}^{c,k}$ (i.e. $u_{n,t}^{c,k}$ is the edge unit). It is trained by minimizing the sum of edge unit prediction and edge duration prediction losses:

$$\mathcal{L}_{uLM} = \sum_{n=1}^{N} (\mathcal{L}_{EU}^n + \mathcal{L}_{ED}^n) \tag{3}$$

$$\mathcal{L}_{EU}^n = \sum_{c=1}^{2} \sum_{k=1}^{2} \sum_{\substack{t \\ u_{n,t}^{c,k} \neq u_{n,t-1}^{c,k}}} \log P(u_{n,t}^{c,k} | u_{n,1:t-1}^{*,k}; \Lambda, \Theta) \tag{4}$$

$$\mathcal{L}_{ED}^n = \sum_{c=1}^{2} \sum_{k=1}^{2} \sum_{\substack{t \\ u_{n,t}^{c,k} \neq u_{n,t-1}^{c,k}}} \left| d_{n,t}^{c,k} - \hat{d}_{n,t}^{c,k}(u_{n,1:t}^{*,k}; \Lambda, \Theta) \right| \tag{5}$$

where $N$ is the total number of utterances in a dialogue, $\hat{d}_{n,t}^{c,k}$ is the continuous duration prediction, and $\Lambda, \Theta$ are prefix tokens and model parameters, respectively.

A.3 INFERENCE PROCEDURE

An example of MS-DLM inference steps, where the total number of utterances $N = 3$ and the context length $C = 4$, is illustrated in Figure A.2. The inference proceeds as follows:

$n = 1$    The phonemes of the first and second sentences are obtained using a grapheme-to-phoneme tool. Since the first sentence will be uttered by speaker A, its phonemes on channel 2 are replaced with LIS tokens. Similarly, the phonemes of second sentence on channel 1 are replaced with LIS tokens. The prefix tokens are then prepared by combining these phonemes with speaker IDs and some special tokens. Note that the context units may be absent or shorter than the context length $C$ for small $n$. The content and pitch units of the first utterance are generated in an autoregressive manner until the EOS token is selected as the content unit for any channel.

$n = 2$    The phonemes of the second and third sentences are obtained in the same manner. Prefix tokens are prepared by incorporating the units generated at $n = 1$ as context units. Then, the content and pitch units of the second utterance are similarly generated.

$n = 3$    Since the inference ends at $n = 3$, phonemes of the $n + 1$th sentence is not used in this step. The content and pitch units generated in the second step are used as context units. Note that since the context length $C = 4$ exceeds the length of units from the previous step (only three units are generated in $n = 2$), context units are additionally derived from the previous context units.

Each step does not rely on input sentences that extend beyond two sentences ahead. For instance, the generation of the first utterance does not rely on the third sentence. This feature facilitates continuous spoken dialogue generation when integrated with an LLM.

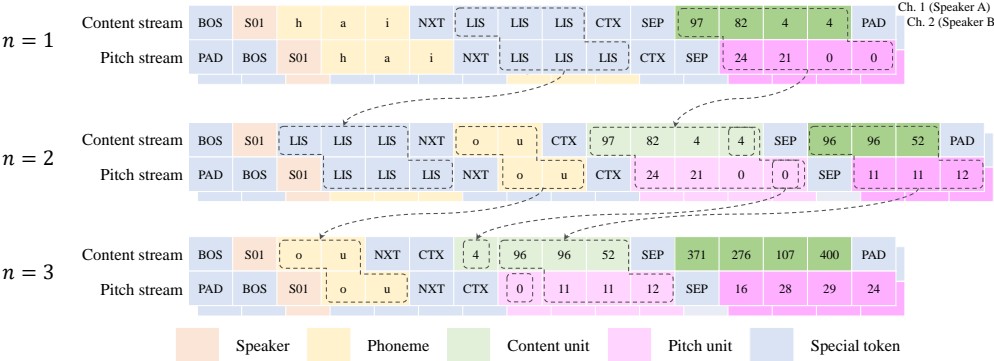

Figure A.2: Conceptual diagram of inference steps with total number of utterances $N = 3$ and context length $C = 4$.

# B EXPERIMENTAL SETUP DETAILS

## B.1 DATASET AND PREPROCESSING

We collected audio recordings of 74 h comprising 538 dialogues conducted by 32 pairs with 54 Japanese speakers (certain speakers appeared in multiple pairs). These dialogues were divided into 474/32/32 for train/valid/test sets, respectively (valid and test sets included all speaker pairs). For the recording sessions, two speakers entered separate soundproof rooms, where they could see and hear each other through glass and via headphones, respectively. Conversations occurred freely and captured in two-channel 96 kHz/24 bit audio.

The recorded 538 dialogues yielded $538 \times 2 = 1,076$ audio files, which were downsampled to 16 and 24 kHz for the s2u and u2s modules, respectively. To eliminate volume discrepancies between different channels and speaker pairs, we calculated the average dBFS of each audio file, and used these averages to normalize the volume levels. Subsequently, the Silero VAD[4] was employed for voice activity detection. Further, we utilized the large model of whisper[5](Radford et al., 2023) for automatic speech recognition on the detected speech segments. Manual corrections for start times, end times, and transcriptions were made for 645 of 1,076 files. Transcripts were automatically converted into phonemes using Open JTalk[6].

## B.2 MODEL, TRAINING, AND INFERENCE

**IPU Classifier:** For the IPU classification task, we employed a 3-layer bidirectional LSTM with the input embedding and hidden dimensions of 256 and 512, respectively. Training was conducted on a single A100 80GB GPU with a batch size of 8,192 tokens, using the Adam optimizer (Kingma & Ba, 2015) with an initial learning rate of $1 \times 10^{-4}$ and betas of $\beta_1 = 0.9$ and $\beta_2 = 0.98$. Our training set comprised 49,339 *s-IPU*s and 27,794 *l-IPU*s, and the model was trained over 20k steps. The checkpoint with the lowest validation loss was selected for final use. When tested on an evaluation set containing 2,604 *s-IPU*s and 1,930 *l-IPU*s, our classifier achieved an accuracy of 87.83%.

**s2u module:** For the s2u module, we used japanese-hubert-base[7] model, a pre-trained HuBERT base model trained on 19k h of Japanese speech, as a frontend for the content unit extractor. It encodes 16 kHz speech into 768-dimensional continuous vectors at 50 Hz. The k-means++ (Arthur & Vassilvitskii, 2007) clustering model was trained on our spoken dialogue dataset described in appendix B.1. In line with Nguyen et al. (2023), the number of clusters was set to 500. The number of bins for pitch unit extraction was 32, one of which was designated for unvoiced frames. The WORLD vocoder (Morise et al., 2016) was used to extract pitch every 20 ms, yielding pitch units at 50 Hz.

**uLM:** We adopted the same hyperparameters as described by Nguyen et al. (2023), utilizing a Transformer model comprising 6 layers, 4 of which were cross-attention layers, with 8 attention heads per layer and an embedding size of 512. The context length $C$ was 500, corresponding to a 10-s waveform. The uLM's vocabulary included 500 content units (with 32 shared with pitch units), 39 phonemes, 9 special tokens, and a combined total of 3,298 speaker IDs (comprising $54 + 3,244$ entries). Special tokens included BOS, EOS, PAD, NXT, CTX, SEP, LIS, as described in section 3.1.2, UNK for unknown input, and LAU for explicitly including laughter in the phoneme sequences. However, outputs are limited to the content/pitch units, PAD, and EOS tokens by setting the output probabilities for other tokens to zero.

A single-channel variant of our uLM was pre-trained on the CSJ dataset, where we simplified the prefix tokens by omitting the phonemes of the next utterance and context units. The refined prefix tokens took the following form:

$$\text{BOS}, s^c, p_{n,1}^c, \ldots, p_{n,M_n}^c, \text{SEP}. \tag{6}$$

---

[4]`https://github.com/snakers4/silero-vad`
[5]`https://github.com/openai/whisper`
[6]`https://open-jtalk.sourceforge.net/`
[7]`https://huggingface.co/rinna/japanese-hubert-base`

Consequently, this phase of pre-training can be regarded as a conventional text-to-speech training. This pre-training employed two A100 80GB GPUs, each managing a batch size of 30,000 tokens. Optimization was performed over 100k steps using an Adam optimizer (Kingma & Ba, 2015) with an inverse square root learning rate schedule, whose initial learning rate was set to $1 \times 10^{-7}$, warmup steps to 10k steps, and maximum learning rate to $5 \times 10^{-4}$. This required approximately 5 h.

Subsequently, we finetuned a two-channel uLM on all of the *s-IPU*s present in our spoken dialogue dataset, which contained 82,060 utterances. As our uLM shares the weight across two Transformer towers, two-channel uLM were warm-started with the pre-trained single-channel uLM weights. Finetuning was conducted in the same configuration as pre-training; however, the maximum learning rate was $1 \times 10^{-4}$, requiring approximately 11 h.

For decoding, we adopted nucleus sampling (Holtzman et al., 2020) with $p = 0.9$. Through empirical observation, we discerned that the top-20 sampling, as utilized for dGSLM (Nguyen et al., 2023), produced speech signals misaligned with the input phonemes. This misalignment likely stems from units with marginally lower probabilities, such as the top-19 or top-20 units, correlating with pronunciations incongruent with the desired phoneme.

**u2s module:** Our u2s module received a global speaker ID with 50 Hz content and pitch units. These discrete values were embedded into 128-dimensional continuous vectors, which were then summed to produce 50 Hz input features. These features were subsequently upsampled by factors of $[10, 6, 4, 2]$ to obtain a 24 kHz waveform. Following Kong et al. (2020), we trained our u2s module with the Adam optimizer, setting an initial learning rate to $2 \times 10^{-4}$ and betas at $\beta_1 = 0.8$ and $\beta_2 = 0.99$. The model was optimized over 500k steps on a single A100 80GB GPU with a batch size of 16 0.5-second speech segments, requiring approximately 32 h. Our training set consisted all of the VAD speech segments from our spoken dialogue dataset, totalling 130,050 utterances. During inference, we decoded the waveform for each channel and utterance individually, as excessive GPU memory would be required to process the entire 5–10 minute dialogue at once.

## C  EFFECTS OF INTRODUCING PITCH UNITS

To explore the effect of the pitch units, we calculated PER for systems without pitch units in the same manner as described in section 4.2. Additionally, we extracted $F_0$ values from the generated speech using the WORLD vocoder, calculated the mean and variance of the voiced frames, and averaged them across all utterances. The results are summarized in Table C.1. Interestingly, the removal of pitch units worsened the PER for *Resynthesized*, whereas it improved the PER for *Baseline* and *Proposed* systems. Thus, the requirement to predict the pitch units rendered it difficult to predict the accurate pronunciation, which is mostly determined by the content units. However, the $F_0$ statistics of systems with pitch units were consistently closer to those of *Ground Truth* than their pitch-ablated counterparts, indicating that the pitch units were effective for generating expressive speech uttered in spoken dialogues.

Table C.1: PER and pitch statistics measured in TTS setting. The lowest PER and $F_0$ statistics closest to the Ground Truth in each section are highlighted in bold.

| METHOD | PER ↓ | $F_0$ mean [Hz] | $F_0$ var [Hz²] |
|---|---|---|---|
| *Ground Truth* | 8.95 | 191.6 | 2831.6 |
| *Resynthesized* | **11.49** | **189.2** | **2509.8** |
| *w/o pitch units* | 12.20 | 177.0 | 2202.8 |
| *Baseline* | 12.13 | **181.8** | **2271.1** |
| *w/o pitch units* | **11.61** | 173.7 | 1802.5 |
| *Proposed* | 13.03 | **186.2** | **2639.4** |
| *w/o pitch units* | **11.17** | 178.1 | 2234.4 |

# D  BACKCHANNEL AND LAUGHTER EVALUATION

## D.1  BACKCHANNEL CONTENT EVALUATION

We transcribed all backchannels in the Ground Truth and generated spoken dialogues using the large model of whisper (Radford et al., 2023). Subsequently, we removed the trailing symbols ("!", "...", "。", etc.) and sorted them by their frequency. The results are shown in Table D.1. These results indicate that our system is capable of appropriately generating backchannels used in actual conversations.

Table D.1: Top-20 frequently used backchannels in Ground Truth and generated spoken dialogues. Each Japanese transcripts were translated into English to match the meaning as closely as possible.

| | Ground Truth | | | | Proposed (Generated) | | |
|---|---|---|---|---|---|---|---|
| Freq. | Transcript | Pronunciation | Translation | Freq. | Transcript | Pronunciation | Translation |
| 261 | うん | un | Uh-huh | 148 | うん | un | Uh-huh |
| 87 | んー | nn | Mm-hm | 117 | ん | n | Mm |
| 58 | はい | hai | Yes | 77 | んんん | nnn | Mmm |
| 47 | そう | sou | I see | 62 | んんっ | nn | Mm! |
| 43 | んんん | nnn | Mmm | 26 | んんんん | nnnn | Mm-hmm |
| 43 | ん | n | Mm | 25 | んん | nn | Mm-mm |
| 32 | うんうん | unun | Yeah yeah | 24 | んー | nn | Mm-hm |
| 25 | んんんん | nnnn | Mm-mm | 24 | はい | hai | Yes |
| 23 | あーー | aaa | Ah | 14 | ふぅ | fuu | (sigh) |
| 21 | うーん | uun | Hmm | 12 | そう | sou | I see |
| 20 | www | (laugh) | (laugh) | 11 | はいはい | haihai | Yes yes |
| 17 | はぁ | ha | Oh | 11 | うんうん | unun | Yeah yeah |
| 16 | そうそうそう | sousousou | Exactly | 10 | あ、そうなんだ | a, sounanda | Oh, is that so? |
| 14 | ふふっ | fufu | (chuckle) | 8 | フフフフフフ | fufufufufu | (laugh) |
| 11 | ねえ | nee | Hey | 7 | はぁ | ha | (sigh) |
| 11 | wwww | (laugh) | (laugh) | 6 | そうそうそう | sousousou | Exactly |
| 11 | んんっ | nn | Mm! | 6 | そうなんだ | sounanda | Oh, really? |
| 10 | んふふふ | nfufufu | (giggle) | 6 | んふふふ | nfufufu | (giggle) |
| 9 | はいはいはい | haihaihai | Yes yes yes | 6 | そうだね | soudane | That's right |
| 9 | んーー | nnn | Mm-hmm | 5 | www | (laugh) | (laugh) |

## D.2  SPEAKER-SPECIFIC CHARACTERISTICS OF BACKCHANNELS

While the overall frequency of backchannels is summarized in Table 2, it actually varies from speaker to speaker. To further probe the speaker characteristics, we computed the proportion of backchannels $100 \times q_{BC}/q_{ALL}$ for each speaker. The mean absolute error (MAE) and Pearson correlation coefficient $r$ between the *Ground Truth* and generated dialogues were calculated. The results are listed in Table D.2. *Proposed* achieved the lowest MAE and exhibited a positive correlation with *Ground Truth*. These results demonstrate that the proposed system can produce backchannels in appropriate frequency, and the speaker characteristics are preserved in the generated spoken dialogues.

Table D.2: Detailed comparison of backchannel frequency for individual speakers between the reference and generated dialogues. Values closest to the *Ground Truth* are bolded. Significance levels of $r$ are shown by $^\dagger$($^\ddagger p < 0.01$, $^\dagger p < 0.05$).

| METHOD | MAE $\downarrow$ | $r \uparrow$ |
|---|---|---|
| *Ground Truth* | 0.00 | 1.00$^\ddagger$ |
| *dGSLM* | 0.09 | **0.63**$^\ddagger$ |
| *Baseline* | 0.18 | 0.40$^\ddagger$ |
| *Proposed* | **0.07** | 0.54$^\ddagger$ |
| *w/o TTM* | 0.14 | 0.54$^\ddagger$ |

### D.3 LAUGHTER EVALUATION

We applied an open-source laughter detection model[8] (Gillick et al., 2021) to the generated spoken dialogues. We then counted the instances of laughter and calculated their total duration. The results are summarized in Table D.3. The frequency and duration of laughter generated by the proposed system were closer to those of the *Ground Truth* compared to those of the *Baseline* and *dGSLM* regardless of the existence of a turn-taking mechanism. Note that the *Baseline*, which cannot generate laughter on the listener side, generated a certain amount of laughter because the input written dialogue often contained laughter. *dGSLM* could not utilize such written information, which led to an underestimation of laughter frequency.

Table D.3: Laughter frequency and duration. Values closest to the *Ground Truth* are bolded.

| METHOD | Frequency | Duration |
|---|---|---|
| *Ground Truth* | 1268 | 2975 |
| *dGSLM* | 998 | 2443 |
| *Baseline* | 1011 | 2373 |
| *Proposed* | **1275** | 2810 |
| *w/o TTM* | 1280 | **3010** |

## E SPEAKER-SPECIFIC CHARACTERISTICS OF TURN-TAKING EVENTS

We analyzed the speaker-specific characteristics of turn-taking event durations following the procedure detailed in appendix D.2. For each speaker, we calculated the median durations of the four turn-taking events, IPU, pause, overlap, and gap. The results are presented in Figure E.1 with their regression lines. The values from *dGSLM* and *Proposed* demonstrate positive correlations between the reference and generated dialogues, indicating the preservation of speaker-specific characteristics. Subsequently, we determined the MAE and Pearson's $r$ values between *Ground Truth* and each system. The results are listed in Table E.1. The performance of *Proposed* was consistently superior to *Baseline* and *Proposed w/o TTM*, and it achieved comparable results to *dGSLM*. Moreover, *dGSLM* leveraged 30 s of recorded speech, whereas *Proposed* did not. Therefore, we conclude that the proposed system effectively utilized the speaker information in the prompt tokens, facilitating the reproduction of the general aspects of turn-taking and the specific characteristics of each individual speaker.

Table E.1: Detailed comparison of turn-taking event durations for individual speakers between the reference and generated dialogues. Values closest to the *Ground Truth* are bolded. Significance levels of $r$ are shown by $^\dagger$($^\ddagger p < 0.01$, $^\dagger p < 0.05$).

| METHOD | IPU | | PAUSE | | OVERLAP | | GAP | |
|---|---|---|---|---|---|---|---|---|
| | MAE ↓ | $r$ ↑ | MAE ↓ | $r$ ↑ | MAE ↓ | $r$ ↑ | MAE ↓ | $r$ ↑ |
| *Ground Truth* | 0.00 | $1.00^\ddagger$ | 0.00 | $1.00^\ddagger$ | 0.00 | $1.00^\ddagger$ | 0.00 | $1.00^\ddagger$ |
| *dGSLM* | 0.25 | $0.35^\dagger$ | 0.09 | $\mathbf{0.42^\ddagger}$ | 0.13 | $\mathbf{0.50^\ddagger}$ | **0.06** | $\mathbf{0.42^\ddagger}$ |
| *Baseline* | 1.40 | $0.38^\ddagger$ | 0.14 | 0.16 | 0.32 | 0.04 | 0.33 | 0.01 |
| *Proposed* | **0.24** | $\mathbf{0.63^\ddagger}$ | **0.08** | $0.42^\ddagger$ | **0.10** | $0.42^\ddagger$ | 0.08 | $0.34^\dagger$ |
| *w/o TTM* | 0.34 | $0.52^\ddagger$ | 0.16 | $-0.09$ | 0.11 | $0.35^\ddagger$ | 0.12 | 0.21 |

---

[8] https://github.com/jrgillick/laughter-detection

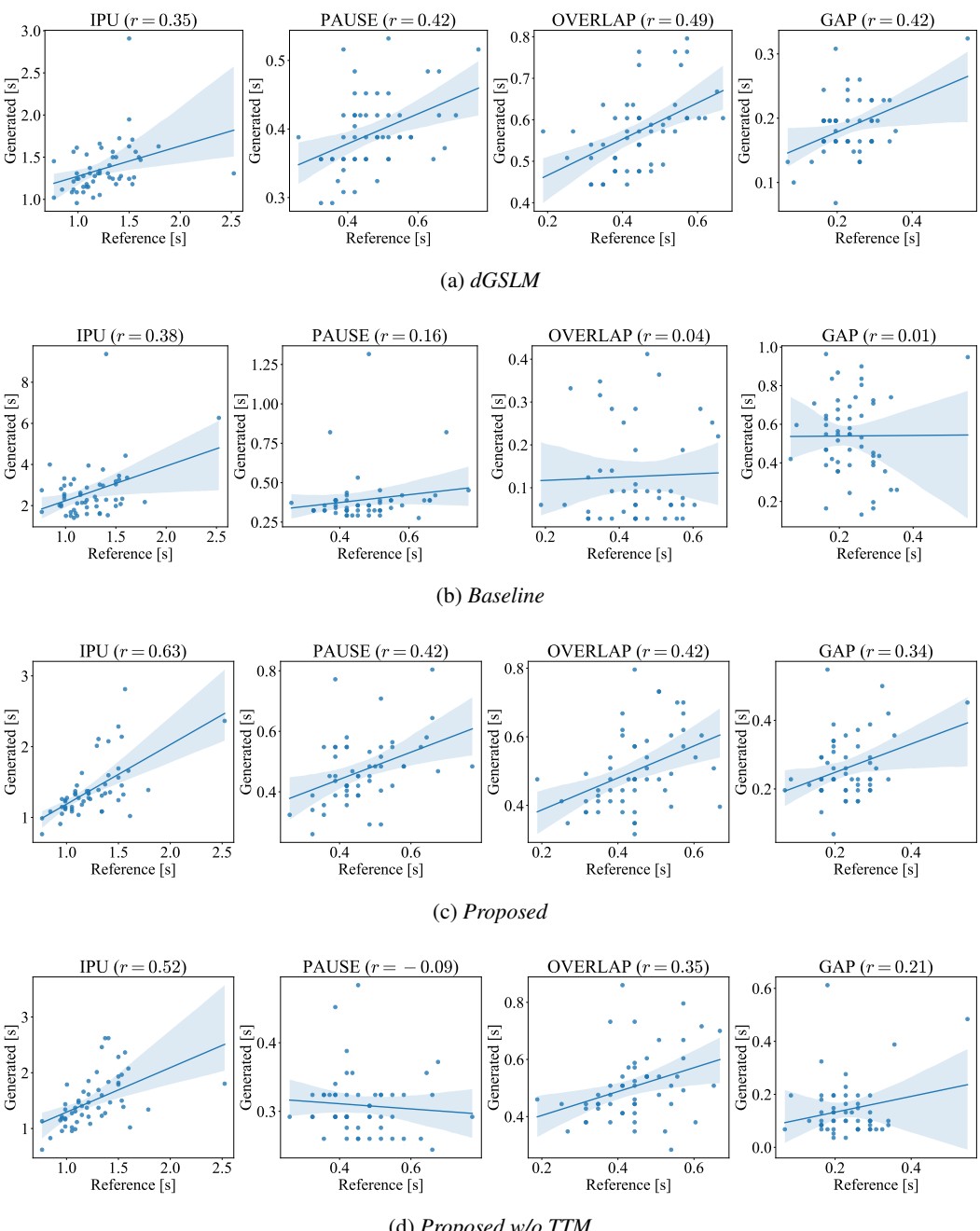

(a) *dGSLM*

(b) *Baseline*

(c) *Proposed*

(d) *Proposed w/o TTM*

Figure E.1: Scatter plot and regression line of the median duration of each speaker's turn-taking events, with the 95% confidence intervals indicated by the shaded region. Each point indicates a different speaker.

## F   HUMAN EVALUATION CRITERIA

For better reproducibility, the instruction used in our human evaluation is presented below. Please note that this instruction is translated from Japanese.

> Please listen to the following audio of friends chatting casually through headphones and evaluate its quality based on these three criteria:
>
> - Dialogue Naturalness: Are backchannels and laughter appropriately included to create a human-like interaction? Is there a seamless transition between the speaker and listener at the right moments? Does the conversation flow smoothly?
> - Meaningfulness: Does the dialogue have meaningful content, and is it possible to understand what is being said?
> - Sound Quality: Is the sound clear and easy to hear, free from noise or other distractions?
>
> Please rate each item on a scale of 1 (bad) to 5 (excellent). When evaluating each criterion, do not consider the other criteria. For example, if the content is incomprehensible but the interaction sounds human-like, rate Dialogue Naturalness highly.

## G   GENERATION CASE STUDIES

We present examples of written dialogues (Table G.1, Table G.2) and the generated spoken dialogues using the proposed system (Figure G.1, Figure G.2). These examples correspond to the test-set sample 1 and 2 of our demo page[9]. Although the original dialogues are in Japanese, we provide their English translation for better readability. As we expected, the entire spoken dialogue closely follows the input written dialogue, with appropriate generation of backchannels and laughter on the listener side. Additionally, some utterances slightly overlap with previous ones, facilitating natural turn-taking. Furthermore, our system can generate laughter on the speaker side by explicitly including a laughter tag (LAU) in the written dialogue, as demonstrated in the sixth segment of Figure G.2. However, upon closer examination of the fourth utterance of Figure G.2, it is observed that the laughter from speaker B is not generated, and instead, the generation of speaker A's utterance begins. This indicates areas for improvement such as ensuring accurate synthesis of the input text content and addressing the issue of too rapid onset of utterance overlap.

Table G.1: The first example of a written dialogue input with utterance index $n$.

| $n$ | Original Script | Translated Script |
|---|---|---|
| 1 | A: 見たりしますね | A: I do watch it. |
| 2 | B: え、すごい、実写かぁ、えっ、エフェクトつける | B: Oh, that's cool, it's live-action, huh, with effects. |
| 3 | B: ってことはあれだよねー、あのー、編集して、実際の | B: So that means, um, editing it, the actual |
| 4 | B: 動きは人間がやって、 | B: movements are done by humans, |
| 5 | B: なんかやってみた感 | B: kind of giving it a try. |
| 6 | A: もうなんかこう、光をこう、ラケットとボールが当たる瞬間にこう入れてみたりとか | A: I just, like, tried adding light, like, at the moment the racket hits the ball, |
| 7 | A: なんかそのー、ボールがそのー、えー、コートに着地した時に、その着地したところが崩れるエフェクトがあって、なんか穴がコートに開くみたいな | A: like, when the ball, um, lands on the court, there's an effect where the landing spot crumbles, like a hole opens up in the court. |
| 8 | B: うわっ | B: Woah |
| 9 | B: そこまでやっちゃうんだ | B: You go that far. |
| 10 | A: そうなんですよ | A: Yes, that's right. |

---

[9]https://anonresearch81.github.io/research/publications/CHATS/

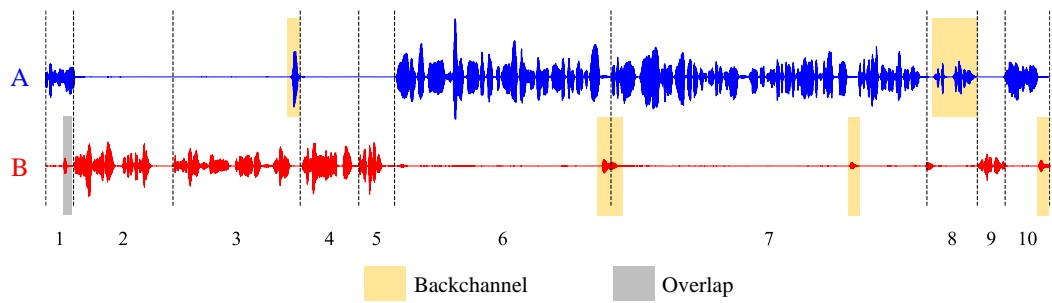

Figure G.1: The first example of a generated spoken dialogue. Dashed lines indicate the boundaries of each utterance, and the numbers from 1 to 10 indicate the indices of the utterances.

Table G.2: The second example of a written dialogue input with utterance index $n$.

| $n$ | Original Script | Translated Script |
|---|---|---|
| 1 | B: なかなかないよね | B: It's pretty rare, isn't it? |
| 2 | A: ふーん、自分で行く、よね、それこそファーストフード | A: Hmm, you'd go there yourself, right, especially for fast food. |
| 3 | A: くらい、よ | A: At least, right. |
| 4 | B: (LAU) | B: (LAU) |
| 5 | A: お安い、回転寿司の方が落ち着くし | A: It's cheaper, and I feel more at ease at conveyor belt sushi places. |
| 6 | A: ねー、いっぱい食べれるしね(LAU)、そうなのよ、結局ね、結局そうなんですよ、結局、そうなる、そこに行くんです | A: Right? You can eat a lot (LAU), exactly, in the end, that's what it comes down to, eventually, that's where we go. |
| 7 | A: やっぱりすごいです | A: It's really amazing. |
| 8 | B: うん、チェーン店は、偉大ということで | B: Yeah, chain stores are, in a sense, remarkable. |
| 9 | B: はい、一旦これで、おわりでいい? | B: Alright, can we conclude this for now? |
| 10 | A: はい、いいですかね | A: Yes, is that okay? |

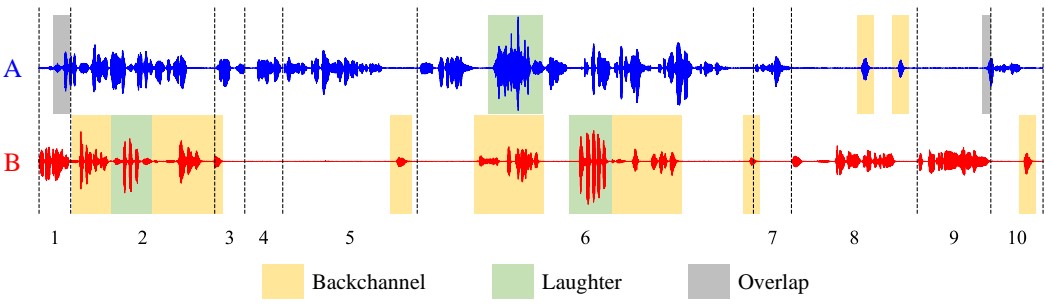

Figure G.2: The second example of a generated spoken dialogue. Dashed lines indicate the boundaries of each utterance, and the numbers from 1 to 10 indicate the indices of the utterances.

