# OpenReview forum: "Towards human-like spoken dialogue generation between AI agents from written dialogue"
_ICLR.cc/2024/Conference — Submitted to ICLR 2024_

### Official Review · Reviewer_Kha2 · 2023-10-27

**Soundness:** 2 fair
**Presentation:** 2 fair
**Contribution:** 3 good
**Rating:** 5
**Confidence:** 3

**Summary:**

This paper proposes CHATS (CHatty Agents Text-to-Speech), a system for transforming written dialogue into spoken dialogue, whose content is coherent with the input written dialogue but generated with backchannels, laughter, and smooth turn-taking. Several contributions are announced: a method to prepare written dialogue by excluding backchannels, a mechanism for taking turns in conversation, and a Multi-Stream Dialogue Transformer Language Model. Paper builds upon previous work, such as dGSLM and Dialogue Transformer Language Model by Nguyen et al. in 2023 and it provides evaluations for different parts of the proposed system, including the dialog model, turn-taking model, and back-channeling model.

When I take a closer look, it's clear that paper has too much stuff in it. There are many models and evaluations crammed together in one document w/o enough details of each of them. This makes it hard to read and understand the paper. It's unfortunate because this is an ambitious and relevant research objective that is described here. Current version of the paper needs a big re-organization to make it clearer and maybe each problem addressed should correspond to a single paper with deeper / more detailed description and evaluation; that would allow reader/reviewer to better understand and appreciate the valuable insights it offers.

**Strengths:**

-This research is ambitious because it explores how people talk in real conversations, not just in written text.

-It introduces a Turn-Taking and Backchanneling Mechanism, which is important for making better autonomous spoken conversational agents.

-The Multi-Stream Dialogue Transformer Language Model (MS-DLM) seems the main contribution and is definitely an interesting architecture

**Weaknesses:**

* It's unfortunate that the spoken dialog examples on GitHub are not in English. This makes it challenging for me to evaluate, and it limits its accessibility since only Japanese speakers can understand it. English examples would have been more universal.

- In the contributions mentioned in the paper, it is not clear why "Conversion from Spoken to Written Dialogue" is valuable or innovative. Authors mention using both rule-based and machine learning approaches to identify backchannels and exclude them from written dialogues, but the paper lacks detail on the challenges this addresses. Is it mainly about data preprocessing?

- As said above, the paper's structure needs improvement, as it tries to cover too many topics in one document.

- The paper builds on the work of dGSLM (Nguyen et al., 2023) and the Dialogue Transformer language model (DLM) (Nguyen et al., 2023), but it doesn't provide enough information about these previous models to make this paper self-understandable

- Section 3.1.2 seems to be a core part, but it's too brief to fully understand its significance.

**Questions:**

see main remarks above +
 not clear was is the challenge in the part "Conversion from Spoken to Written Dialogue"

---

> ### Author Response · Authors · 2023-11-14
> **Response to Reviewer Kha2 (1/2)**
>
> We thank Reviewer Kha2 for the insightful feedback on our paper. We are currently revising our manuscript to incorporate your suggestions, and will inform you once the updated manuscript is ready. The revised version will include (1) a clearer statement on the importance of “Conversion from Spoken to Written Dialogue” in Section 2, (2) a detailed explanation of Generative Spoken Language Model (GSLM) ([Lakhotia et al., 2021](https://arxiv.org/abs/2102.01192)) pipeline in a new section or Appendix, and (3) an expanded discussion on the differences between conventional DLM and our MS-DLM in Section 3 or Appendix. Below are our specific responses to each of your comments.
>
> **Response to Complexity and Organization Concerns**:
> > When I take a closer look, it's clear that paper has too much stuff in it. There are many models and evaluations crammed together in one document w/o enough details of each of them. This makes it hard to read and understand the paper. It's unfortunate because this is an ambitious and relevant research objective that is described here. Current version of the paper needs a big re-organization to make it clearer and maybe each problem addressed should correspond to a single paper with deeper / more detailed description and evaluation; that would allow reader/reviewer to better understand and appreciate the valuable insights it offers.
>
> Thank you for your constructive feedback and the recognition of our research's ambition. Our approach integrates various elements critical to naturalistic spoken dialogue generation, such as backchannels, laughter, and turn-taking. We believe segmenting these elements into separate papers could detract from the integrated nature of these components. To improve readability, we will clarify the conventional GSLM pipeline including s2u/u2s modules and uLM as well as its adaptations in our system, enhancing the manuscript's comprehensibility.
>
> **On Language Accessibility of Examples**:
> > It's unfortunate that the spoken dialog examples on GitHub are not in English. This makes it challenging for me to evaluate, and it limits its accessibility since only Japanese speakers can understand it. English examples would have been more universal.
>
> Thank you for your feedback regarding the language accessibility of our examples. To address your concern, we plan to add figures like Figure E.1 and E.2 to our demo page. These figures will visually indicate the generated backchannels, laughter, and overlaps, making the content more accessible to non-Japanese readers
>
> We acknowledge the importance of including English examples and intend to incorporate them in the future. However, currently, we face a limitation due to the lack of a suitable dataset.
> Our system requires two-channel audio, where the voices of each speaker are separately recorded in each channel. However, such public datasets are currently rare, which led us to decide to record our own high-quality spoken dialogue dataset.
> For example, the Fisher dataset, though used in the previous study, is not ideal for demonstrating our system's capability of high-quality spoken dialogue generation, due to its limited bandwidth (8 kHz sampling rate, 4kHz bandwidth).
>
> In addition, we would like to highlight our system’s language-independent design.
> Although backchannels and laughter vary across languages (e.g. "un", "hai" in Japanese, "uh-huh", "yeah" in English, and "shi", "dui" in Mandarin), our system learns the pronunciation and timing of them in a data-driven manner using language-independent "units". Therefore, we believe our system is also applicable to languages other than Japanese. More specifically, we just have to replace the grapheme-to-phoneme module with one tailored to the target language to obtain the correct phoneme sequence $p_{n,1}^c, …, p_{n, M_n}^c$.

---

> > ### Author Response · Authors · 2023-11-14
> > **Response to Reviewer Kha2 (2/2)**
> >
> > **Clarifying “Conversion from Spoken to Written Dialogue”**:
> > > In the contributions mentioned in the paper, it is not clear why "Conversion from Spoken to Written Dialogue" is valuable or innovative. Authors mention using both rule-based and machine learning approaches to identify backchannels and exclude them from written dialogues, but the paper lacks detail on the challenges this addresses. Is it mainly about data preprocessing?
> >
> > Thank you for highlighting the need for clarity regarding the "Conversion from Spoken to Written Dialogue" in our paper. Your feedback helped us realize that we did not sufficiently explain its significance in our methodology. This conversion process is integral to preparing input texts for training our CHATS system. Let's consider a scenario where we have a two-channel audio recording of a spoken dialogue between two participants. We can transcribe this dialogue using automatic speech recognition or manual annotation (as depicted in Figure 1(a)). However, this raw transcription will include elements characteristic of spoken language, such as backchannels and laughter, which are typically absent in standard written dialogues (refer to Figure 1(b)).
> >
> > Training the CHATS system directly on these raw transcriptions would be suboptimal, as the system might then fail to generate these spontaneous spoken behaviors when given a typical written dialogue as input. To address this, we actively remove elements like backchannels and laughter from the transcriptions. This modification ensures that the system learns to autonomously generate these behaviors in the listener's responses, aligning more closely with natural spoken communication patterns. This step is crucial for the effective functioning of the CHATS system, bridging the gap between written and spoken dialogues.
> >
> > **On GSLM and DLM Background**:
> > > The paper builds on the work of dGSLM (Nguyen et al., 2023) and the Dialogue Transformer language model (DLM) (Nguyen et al., 2023), but it doesn't provide enough information about these previous models to make this paper self-understandable
> >
> > We recognize the need for more background information on GSLM and DLM. The revised manuscript will include a comprehensive overview of these models, aiding in understanding our system’s architecture and its advancements.
> >
> > **Elaborating on Section 3.1.2**:
> > > Section 3.1.2 seems to be a core part, but it's too brief to fully understand its significance.
> >
> > We think this also owes to the lack of background explanation. We will clarify the difference between the conventional DLM and proposed MS-DLM in terms of model architecture, design of input/output sequences, and training objective.

---

> > > ### Comment · Reviewer_Kha2 · 2023-11-19
> > > **after authors' answers**
> > >
> > > tks for your answers, i acknowledge authors' amibition to improve paper structure / readability and add description on dGSLM in the background +  as well as providing english samples on their demo page; i'll slightly increase my score accordingly

---

> > > > ### Author Response · Authors · 2023-11-20
> > > > **Additional Response to Reviewer Kha2**
> > > >
> > > > Thank you for acknowledging our efforts to enhance the paper's structure, readability, and the addition of dGSLM descriptions in the background section.
> > > > In response to your comments, we have carefully revised the manuscript and incorporated the suggested amendments. We believe these changes have further improved the clarity and quality of our work.
> > > >
> > > > We have uploaded the updated manuscript for your review. If your schedule permits, we would be grateful for any further insights or comments you might have on the revised version.
> > > >
> > > > Thank you once again for your constructive feedback and for the opportunity to refine our paper.

---

### Official Review · Reviewer_yc8m · 2023-11-04

**Soundness:** 4 excellent
**Presentation:** 3 good
**Contribution:** 3 good
**Rating:** 8
**Confidence:** 4

**Summary:**

The paper proposes a method to generate natural overlapping spoken dialogue with the listener cues like backchannels and laughter only using the written transcripts (that lack the rich spoken dialog modes). This system generates speech for both the speaker and the listener simultaneously, using only the transcription from the speaker side by finetuning the modified dGSLM model with careful curation and pre-processing of natural dialog. The overall pipeline is similar to the one used by the dGSLM system; however using the careful finetuning process delivers very strong results and a practical tool for enriching the dialogs with natural spoken dialog properties.

The model has extensive experiments to show that the utterance quality is good, the dialog segments contain high quality of close to ground truth backchannels and pauses and the turn taking events also resemble the ground truth. Most important, the qualitative human evaluation experiments also show very good naturalness, meaningfulness and sound quality.

**Strengths:**

- There are many Dialog generation LLMs available today. These are currently not very natural generation systems, meaning, they cannot mimic human-to-human conversations that contain rich elements like laughter, backchannel, fluid turn-taking, etc. This paper aims to solve this problem and generates natural spoken dialog and presents methods including how to prepare datasets, create context properly in the training data and predict turn-taking events using the dual-transformer architecture (dGSLM).
- The methods also shows how smaller datasets (74 hr of 2 channel speech) can be used to train a high quality spoken dialog generator (using a pre-trained uLM model).
- Ablations show that data augmentation, next sentence objectives, turn-taking mechanism were all important pieces of the architecture and pipeline are all important for getting the overall natural dialog output.

**Weaknesses:**

- the paper presents the overall system very well, however, it is not clear if the original contribution of the work is significant. It seems like a straightforward extension of the dGSLM model where it has been fine-tuned to create this improved version of natural dialog corpora.
- there is no comparison to any other baseline system that is described in the paper.
- human evaluation does not try to assess the content and quality of generated backchannels.
- Also, it is not clear how the generation will transfer to various other data domains.

**Questions:**

- it is not clear how many backchannel tokens are in the vocabulary (like laughter, ums, etc).

---

> ### Author Response · Authors · 2023-11-15
> **Response to Reviewer yc8m (1/3)**
>
> We thank Reviewer yc8m for the insightful comments regarding our paper. We are in the process of revising our manuscript to address the concerns raised and will inform you once the updated manuscript is ready. Below, we provide our specific responses to each of your comments.
>
> **On Summary and Strengths**:
>
> We deeply appreciate your insightful and precise summary of our research's novelty and strengths. Your acknowledgment of the significant aspects and contributions of our work is invaluable and reinforces the importance of our efforts in advancing spoken dialogue generation. Thank you for your thorough and thoughtful analysis.
>
> **On Original Contribution and Significance**:
>
> > the paper presents the overall system very well, however, it is not clear if the original contribution of the work is significant. It seems like a straightforward extension of the dGSLM model where it has been fine-tuned to create this improved version of natural dialog corpora.
>
> As you rightly pointed out, our CHATS system builds upon the foundations of existing models such as GSLM([Lakhotia et al., 2021](https://arxiv.org/abs/2102.01192)) and dGSLM([Nguyen et al., 2023](https://arxiv.org/abs/2203.16502)).
> However, it is crucial to recognize that the advancement of deep learning often involves the innovative adaptation of powerful existing models for new tasks. Successful examples include VALL-E ([Wang et al., 2023](https://arxiv.org/abs/2301.02111)) and AudioPaLM ([Rubenstein et al., 2023](https://arxiv.org/abs/2306.12925)), which utilize GPT-like Transformer decoder architecture for TTS and ASR/ST, respectively.
>
> CHATS distinguishes itself from dGSLM by offering a unique input/output structure specifically designed for spoken dialogue generation. This design allows for the control of spoken content through textual input and enables the generation of contextually accurate and coherent utterances from written dialogues, a capability absent in dGSLM. This advancement, we believe, is significant both academically and in practical applications across various domains.
>
> **On Lack of Comparative Baseline Systems:**:
>
> > there is no comparison to any other baseline system that is described in the paper.
>
> We appreciate your observation regarding the comparative baselines. Given that our study explores AI-to-AI spoken dialogue generation, a relatively novel area, we faced challenges in identifying appropriate conventional baselines. Traditional TTS systems, including recent state-of-the-art ones, do not incorporate elements like backchannels, laughter, or turn-taking management. Therefore, we chose a standard TTS system whose architecture is similar to the proposed system and dGSLM as our baselines to cover a spectrum from text-driven to textless approaches. Our focus was on conducting in-depth analyses of key components through ablation studies, which we believe will more effectively advance the field of spoken dialogue generation.

---

> ### Author Response · Authors · 2023-11-15
> **Response to Reviewer yc8m (2/3)**
>
> **On Evaluation of Generated Backchannels:**
>
> > human evaluation does not try to assess the content and quality of generated backchannels.
>
> We recognize the importance of evaluating the content and quality of generated backchannels and appreciate your suggestion. In fact, we included the naturalness of backchannels as one of the criterion of dialogue naturalness evaluation as follows:
> *"Dialogue Naturalness: Are backchannel and laughter appropriately integrated to create a human-like interaction? Is there a seamless transition between the speaker and listener at the right moments? Does the conversation flow smoothly?"*
> Therefore, we conjecture that if the backchannel content was unnatural or its quality was quite low, the dialogue naturalness score of the Proposed system could be lower than the Baseline system in Table 5. For greater clarity and reproducibility, we will include the exact wording used in our human evaluation in the Appendix.
>
> Additionally, since we segmented the generated backchannels from the entire spoken dialogue in Section 4.3.1, we applied Whisper to each segment to obtain their transcription and listed the top-20 frequently used backchannels (Please refer to the two tables below for the results). While the order and frequency differ, many backchannels are common between the Ground Truth dialogue and the generated ones, indicating that the CHATS system is capable of appropriately generating backchannels used in actual conversations. We plan to add these results in Appendix.
>
> Table: Top-20 Frequently Used Backchannels in Ground Truth (left four columns) and Generated (right four columns) Spoken Dialogues. Each Japanese transcripts were translated into English to match the meaning as closely as possible.
>
> | Frequency | Transcript | Pronunciation | Translation | Frequency | Transcript | Pronunciation | Translation |
> |-------|------------|---------------|-------------|-------|------------|---------------|-------------|
> | 261 | うん | un | Uh-huh | 148 | うん | un | Uh-huh |
> | 87 | んー | n- | Mm-hm | 117 | ん | n | Mm |
> | 58 | はい | hai | Yes | 77 | んんん | nnn | Mmm |
> | 47 | そう | sou | I see | 62 | んんっ | nn | Mm! |
> | 43 | んんん | nnn | Mmm | 26 | んんんん | nnnn | Mm-hmm |
> | 43 | ん | n | Mm | 25 | んん | nn | Mm-mm |
> | 32 | うんうん | unun | Yeah yeah | 24 | んー | n- | Mm-hm |
> | 25 | んんんん | nnnn | Mm-mm | 24 | はい | hai | Yes |
> | 23 | あーー | a-- | Ah | 14 | ふぅ | fuu | (sigh) |
> | 21 | うーん | u-n | Hmm | 12 | そう | sou | I see |
> | 20 | www | (laugh) | (laugh) | 11 | はいはい | haihai | Yes yes |
> | 17 | はぁ | ha | Oh | 11 | うんうん | unun | Yeah yeah |
> | 16 | そうそうそう | sousousou | Exactly | 10 | あ、そうなんだ | a, sounanda | Oh, is that so? |
> | 14 | ふふっ | fufu | (chuckle) | 8 | フフフフフフフ | fufufufufufu | (laugh) |
> | 11 | ねえ | nee | Hey | 7 | はぁ | ha | (sigh) |
> | 11 | wwww | (laugh) | (laugh) | 6 | そうそうそう | sousousou | Exactly |
> | 11 | んんっ | nn | Mm! | 6 | そうなんだ | sounanda | Oh, really? |
> | 10 | んふふふ | nfufufu | (giggle) | 6 | んふふふ | nfufufu | (giggle) |
> | 9 | はいはいはい | haihaihai | Yes yes yes | 6 | そうだね | soudane | That's right |
> | 9 | んーー | n-- | Mm-hmm | 5 | www | (laugh) | (laugh) |
>
> **On Applicability to Various Data Domains**:
>
> > Also, it is not clear how the generation will transfer to various other data domains.
>
> We understand that the transferability of our model to diverse data domains is an important aspect of its applicability.
> Although we would like to apply our system to languages other than Japanese (e.g. English, Mandarin, etc.) and domains other than chit-chat (e.g. interview, consulting, etc.), we leave it as future work due to the lack of a suitable dataset. Our system requires two-channel audio where each speaker’s voices are separately recorded in each channel, but such a public dataset is rare for now. For example, the Fisher dataset, used in the previous study, is not ideal for demonstrating our system's capability of high-quality spoken dialogue generation, due to its limited bandwidth (8 kHz sampling rate, 4kHz bandwidth).
> This limitation prompted us to begin our research by recording a high-quality spoken dialogue dataset.
>
> In addition, we would like to emphasize that our system learns when and how to generate backchannels or laughter in a data-driven manner by using "units".
> Therefore, we believe that our system is applicable to other languages by simply substituting the grapheme-to-phoneme module. Conditioning our system on domain information to control the content and frequency of backchannels and laughter might also present an interesting research direction."

---

> ### Author Response · Authors · 2023-11-15
> **Response to Reviewer yc8m (3/3)**
>
> **On Vocabulary of Backchannel Tokens**:
>
> > it is not clear how many backchannel tokens are in the vocabulary (like laughter, ums, etc).
>
> We apologize for any confusion caused regarding the backchannel tokens in our vocabulary. To clarify, our vocabulary does not contain specific tokens for each type of backchannel, such as "umm" or "uh-huh". Instead, all backchannels and laughter from the listener side are generated using a single special token, "LIS", as described in Section 3.1.2. This approach allows for a more dynamic and contextually appropriate generation of listener responses without being constrained by a predefined set of backchannel types.
>
> We have listed all special tokens in the Appendix A.2.
> The inclusion of the "LAU" token in our vocabulary is specifically for cases where explicit laughter generation is desired on the speaker side. An example of this can be seen in Table E.2 of Appendix E, where the "LAU" token is used to generate laughter in the speaker side. This method provides a balance between the flexibility of spontaneous listener interactions and the ability to explicitly control certain aspects of the dialogue, like speaker-initiated laughter.

---

### Official Review · Reviewer_FEtd · 2023-11-10

**Soundness:** 4 excellent
**Presentation:** 3 good
**Contribution:** 2 fair
**Rating:** 6
**Confidence:** 4

**Summary:**

This paper tackles the task of generating spoken dialogues between 2 parties using autoregressive models. It follows the earlier work on DLM (dialogue language model), and tries to extend it for better turn-taking and pause modeling. This results in more natural generated dialogs. The authors train all models from scratch.

**Strengths:**

It makes sense to incorporate pitch and content units in a multi-stream dialog language model for spoken dialog generation. The authors also build secondary models for turn taking and pause modeling. These are very critical for a more natural sounding dialog generation, and are lacking in textual dialogs. Especially the audio samples with overlapping speech are impressive.

**Weaknesses:**

I had a hard time to understand the concept of "units" and has to read Kharitonov. The paper should do a better job explaining what they are with motivation. Furthermore I had a hard time understanding uLM and had to read the DLM paper. The authors should first explain DLM. But after reading these 2 papers, it is clear that the contribution is actually not that significant, but still very creative idea, applied to Japanese data.

**Questions:**

dGSLM is trained with 2000 hours of English data. In this paper authors use only 74 hours of Japanese data. And they train the dGSLM models from scratch using that 74 hours. The experimental results show inferior comprehensiveness compared to the original dGSLM paper. This begs the question of authors replicating the experiments for English with larger training set. In other words, we do not know whether their improvements will disappear with more data.

---

> ### Author Response · Authors · 2023-11-14
> **Response to Reviewer FEtd**
>
> We thank Reviewer FEtd for the insightful comments and questions regarding our paper. We are in the process of revising our manuscript to incorporate your suggestions and will inform you once the updated manuscript is ready. Below are our specific responses to each of your comments.
>
> **On Need for Better Explanation of Units and DLM**:
>
> > I had a hard time to understand the concept of "units" and has to read Kharitonov. The paper should do a better job explaining what they are with motivation. Furthermore I had a hard time understanding uLM and had to read the DLM paper. The authors should first explain DLM.
>
> We acknowledge that the original manuscript lacks sufficient explanation of the necessary background knowledge. To rectify this, we will include a more thorough explanation of the Generative Spoken Language Model (GSLM) ([Lakhotia et al., 2021](https://arxiv.org/abs/2102.01192)), which will provide foundational knowledge on "units" and the pipeline of speech-to-unit (s2u) module, unit language model (uLM), and unit-to-speech (u2s) module. Additionally, in Section 3, we will summarize the original DLM ([Nguyen et al., 2023](https://arxiv.org/abs/2203.16502)) and discuss how it differs from our MS-DLM.
>
> **On Significance of Contribution and Creativity**:
>
> > But after reading these 2 papers, it is clear that the contribution is actually not that significant, but still very creative idea, applied to Japanese data.
>
> We extend our sincere gratitude to the reviewer for the effort in reading the cited papers and acknowledging the creative aspect of our work.
> While we agree that our paper builds upon existing models, we believe our contribution lies in the integration of multiple spoken dialogue characteristics such as backchannels, laughter, and turn-taking. These characteristics have typically been addressed separately in previous research.
>
> While we currently focus on Japanese data, we believe that our system is adaptable to other languages due to the language-independent design of the GSLM pipeline.
> Concretely, we just have to substitute the grapheme-to-phoneme module with that of the target language. We plan to extend our system to additional languages as suitable datasets become available.
>
> **On Comparison with dGSLM and Data Volume**:
>
> > dGSLM is trained with 2000 hours of English data. In this paper authors use only 74 hours of Japanese data. And they train the dGSLM models from scratch using that 74 hours. The experimental results show inferior comprehensiveness compared to the original dGSLM paper. This begs the question of authors replicating the experiments for English with larger training set. In other words, we do not know whether their improvements will disappear with more data.
>
> This observation is quite perceptive. Although we are currently unable to conduct additional experiments using the Fisher dataset, we can refer to the original dGSLM paper, which reported that the dGSLM's meaningfulness score was 2.48±0.49, while the Ground Truth conversation achieved 4.21±0.25. These results suggest that even with 2,000 hours of data, it is challenging for the original dGSLM to generate comprehensible content. We hypothesize that a larger dataset narrow the gap between the meaningfulness of dGSLM and our system. However, we believe our system will remain valuable with a massive dataset because it allows for controlled spoken content through text.

---

### Author Response · Authors · 2023-11-20
**For All Reviewers: Paper Revision**

We again thank all reviewers for their invaluable feedback.
In response, we have substantially reorganized the paper to enhance its clarity and self-explanatory nature, as well as to distinguish it more clearly from previous works.

The revised version of the paper is now available, with newly added or modified contents highlighted in blue.
Below is a summary of the revisions:

**[Section 1] Contribution**:
We previously presented the conversion from spoken dialogue transcription to written dialogue format as a key contribution. While this remains important, we now place greater emphasis on our extensive investigation into the characteristics of generated spoken dialogue.

**[Section 2] Background**:
A new "Background" section has been added, providing a concise overview of the GSLM pipeline and dGSLM, complemented by simple figures. This addition significantly aids in understanding the "CHATS" section.

**[Section 3.1] System Architecture**:
The motivation for introducing content and pitch units is now more clearly articulated, supported by the "Background" section.
The importance of prefix tokens in the uLM section has been underscored, as illustrated in Figure 3 (the boundary between prefix tokens and training target is clearly displayed).
Due to space constraints, we have relocated the training objective formula, similar to that of dGSLM, to Appendix A.2.

**[Section 3.2.1] Written Dialogue Preparation via Backchannel Exclusion**:
This section, formerly part of Section 2, has been integrated with discussions on the "turn-taking mechanism" and "data augmentation by context reduction."
These changes more effectively convey our preprocessing and modeling techniques for spoken dialogue.
The significance of this process for training CHATS is also clarified.

**[Section 4.3] Dialogue-Level Evaluation**:
Owing to space limitations, the investigation of speaker-specific characteristics of backchannels and turn-taking events has been moved to Appendices D.2 and E, respectively.

**[Appendix A] MS-DLM Details**:
In addition to the model architecture and training objective, we have provided a detailed explanation of the inference procedure for better understanding.

**[Appendix D] Backchannel and Laughter Evaluation**:
As mentioned in our response to Reviewer yc8m, we have listed the top-20 frequently used backchannels in both Ground Truth and generated spoken dialogues. This demonstrates the CHATS system's capability to appropriately generate backchannels used in actual conversations. For the benefit of non-Japanese speakers, pronunciations and English translations are provided.

**[Appendix F] Human Evaluation Criteria**:
We realized that the original description of human evaluation criteria in Section 4.4 lacked clarity. To improve this, we have included the exact instructions used in our human evaluations for better clarity and reproducibility.

**[Appendix G] Generation Case Studies**:
We have added the original (Japanese) script to present actual inputs and their nuanced meanings.

In addition to the revisions made to the paper, we have added diagrams to the "LLM samples" section of our [demo page](https://anonresearch81.github.io/research/publications/CHATS/).
These diagrams visually indicate which parts of the generated spoken dialogue correspond to backchannels, laughter, and overlaps.
We hope that these visual aids will facilitate understanding for non-Japanese speakers.

---

### Meta-Review · Area_Chair_ckFG · 2023-12-06

**Metareview:**

This paper presents an approach to generate human-like spoken dialogues from written interactions. While the idea and the work is very interesting, a main concern is the clarity of the presentation in the paper and the organization of the presentation. Authors took the reviewer suggestions positively and tried to address the lack of clarity in their rebuttal, the effort resulted in one reviewer increasing their score. However, given the extent of these suggestions, it would be better to have another full review of this paper before acceptance.

**Justification For Why Not Higher Score:**

Author's seem to address clarity concerns, but given the suggestions, I feel we need to re-read and re-review this paper.

**Justification For Why Not Lower Score:**

N/A

---

### Decision · Program_Chairs · 2024-01-16

Reject